# From Antiblackness to Cultural Health in Higher Education

**Tabitha Grier-Reed \*** , **Roun Said and Miguel Quiñones**

Family Social Science, University of Minnesota, St. Paul, MN 55108, USA; rsaid@umn.edu (R.S.);
quino038@umn.edu (M.Q.)
\* Correspondence: grier001@umn.edu

**Abstract:** Antiblackness has a long and storied history in higher education in the United States, and unfortunately, antiblack attitudes and practices continue in the 21st century. With implications for countering antiblackness in higher education and institutionalizing support for cultural health and wellness, we documented experiences of antiblackness in the African American Student Network (AFAM). AFAM was a weekly networking group, co-facilitated by Black faculty and graduate students, where Black undergraduates could come together and share their experiences. Participation in AFAM was associated with Black holistic wellness, and AFAM was a source of cultural health, where we conceptualized cultural health as having a sense of pride and resilience in one's cultural background. We analyzed notes of 277 AFAM discussions from 2005–2006 to 2017–2018 using an adaptation of consensual qualitative research methods to identify four domains of antiblackness: racial trauma (n = 51), racial microaggressions (n = 34), racial rejection (n = 33), and systemic racism (n = 25). In moving from antiblackness to cultural health, we advocate for institutional resources in higher education, such as an institute for cultural health on campus, that values the cultures of Black students and students of color, and that focuses on building communities in which students can generate a wellspring of pride and resilience in their cultural backgrounds.

**Keywords:** antiblackness; cultural health; African American Student Network (AFAM)

## 1. Introduction

Even though Black students are no longer excluded from predominantly White institutions (PWIs) in the United States (U.S.), these spaces are still steeped in antiblackness. Antiblackness in U.S. education is crystallized by the perception that the Black student is "inherently uneducable, or at very least, unworthy of education" [1] (p. 16). The hostile antiblack climates of PWIs in the U.S. not only impact the intellectual wellbeing of Black students but also affect students' emotional wellbeing, which can result in racial battle fatigue [2]. Answering the call of this "Diversity and Equity in Higher Education" Special Issue to extend antiracism work in higher education, we focus on addressing antiblackness in U.S. higher education, where our research study took place. We seek to provide a pathway from antiblackness in education to cultural health by examining discussions that occurred in the African American Student Network (AFAM) and by centering Black students' experiences (where AFAM was a cultural support developed by two Black professors that met weekly for 15 years, bringing together students over the lunch hour to connect and make meaning of their experiences), including their experiences of antiblackness. We should note that although the particularities of antiblackness in our research are nuanced by the sociohistorical context and culture of the United States, antiblackness in and of itself is not limited to the U.S. Hence, our study may also be of interest to those outside of the U.S.

### 1.1. History of Antiblackness in Higher Education

Antiblackness has a long and storied history in U.S. higher education, where oppressive policies, structural violence, and blatant disregard for Black existence stem from the

White racial frame [3] that is dominant in society. Rooted in antiblackness, the White racial frame "is a set of cultural narratives and symbols" [4] (p. 65) that uphold White racial privilege, positing Whiteness as "more virtuous, intelligent, moral, and honest" [5] (p. 145). The White racial frame provides the foundation for structural racism, where antiblackness encapsulates the continuous dehumanization and marginalization of Black people in the U.S. [6].

Mustaffa [7] traces antiblackness in higher education across three historical eras: Colonial Colleges (1745–1775), New Departures (1850–1890), and the Academic Revolution (1945–1975). In the Colonial Era, Black people were not even recognized as human. Excluded from attending institutions of higher education based on their race, Black people were used as property to build up and serve White institutions [7]. The era of Colonial Colleges shaped antiblack legislation and an antiblack culture [8].

Following Colonial Colleges, the New Departure Era occurred after the Civil War, and at this time, Jim Crow laws legalized segregation, which continued the racial trauma endured by Black people as they were not only denied their human rights, but the laws and policies of the time "subjected them to White terrorism through lynching, dehumanizing de facto and de jure policies, education exclusion, and extreme wage exploitation" [7] (p. 716). During this era, many institutions were receiving land-grant funding from the federal government. However, in a post-Civil War society, Black people were still excluded from attending these institutions due to racist admission policies. This racial rejection led to the creation of Black colleges, but without the lands or funding granted to White institutions, this meant relying on donations [7].

The Academic Revolution era was the start of the fight to end Jim Crow, and change began to occur with the Civil Rights Act of 1964, followed by the Higher Education Act of 1965. While Black students started to gain access to institutions of higher education, the policies were still steeped in White supremacy and antiblackness. This led to a tokenization of accepted Black students and the belief that only a very limited number or exceptional group of Black students were qualified to receive an education. The rationale was that those who were kept out were excluded because of poor qualifications, not racism [7].

## 1.2. Antiblackness in the 21st Century

Unfortunately, antiblack attitudes and practices in higher education continue in the 21st century. For example, in describing White institutional presence at PWIs, Gusa [9] (p. 473) "suggest[ed] that White students view minority students as being illegitimate participants in higher education, believing that Whites are unfairly losing ground to Blacks through affirmative action." Moreover, verbal insults, threats, and violence are ways these antiblack attitudes are enacted on campus [9]. These experiences of racism can be traumatic, resulting in racial trauma. "Racial trauma refers to the events of danger related to real or perceived experience of racial discrimination, threats of harm and injury, and humiliating and shaming events, in addition to witnessing harm to other ethnoracial individuals because of real or perceived racism" [10] (pp. 249–250).

Navigating educational environments steeped in antiblackness and White supremacy can feel like death from a thousand cuts [11], wounding the souls of Black people. Ivey et al. [12] assert that Soul Wounds are a result of historical trauma for Indigenous People and People of Color in the United States, where "African Americans have their own version of the Soul Wound [as] a result of continued individual and institutional racism since the time of slavery and 'Jim Crow' segregation" [12] (p. 35). Racial battle fatigue [13] is a consequence of these Soul Wounds and can include stress, frustration, exhaustion, and depression [14]; however, there is also the possibility for healing.

## 1.3. Conceptualizing Cultural Health

Creating and growing communities focused on wellbeing that foster psychological liberation for Black people is one path toward healing and highlights the need for institutionalized spaces of cultural support or cultural health. Although previous research

has focused on the benefits of collective coping and culturally-specific strategies [15–19], few studies conceptualize cultural health as we do. Exceptions include Grier-Reed and Ajayi [15] and perhaps Grier-Reed, Maples et al. [14] who identify the importance of therapeutic counterspaces for working through racialized labor in ways that buffer racial battle fatigue and connect to psychological liberation in order to achieve cultural health. In contrast, the predominant use of the term cultural health focuses on mental or physical health in different cultural groups and/or health disparities across racial groups.

We conceptualize cultural health in the way of Ivey et al. [12]—as stemming from a sense of pride and resilience in one's cultural background. Achieving and maintaining cultural health in the face of antiblackness is no small feat. What can be considered a labor of love, the labor involved in achieving cultural health as a Black person can aptly be described as racialized labor, a term coined by Grier-Reed, Maples et al. [14]. These authors identified six categories of racialized labor, including the two categories of questioning and affirming [14]. The category of questioning involved "questioning ... [and] critiquing ... [W]hiteness" and the category of affirming involved "affirming and defining Blackness" [14] (p. 112). Both are necessary to develop a sense of pride and resilience in one's cultural background (i.e., cultural health). Moreover, we assert that only institutionalized spaces of cultural support or cultural health will be sufficient to counter the institutionalized nature of antiblackness in education.

*1.4. The African American Student Network (AFAM)*

Institutional resources that promote cultural engagement can facilitate cultural health or knowing oneself and feeling comfortable in one's own Black skin. The African American Student Network (AFAM) where the current research took place has been associated with several aspects of Black holistic wellness [20]. Black holistic wellness is a multidimensional model of wellbeing that includes cultural engagement and communalism, where cultural engagement consists of being involved in cultural activities and knowing one's heritage, and communalism consists of social support and social engagement [20]. Moreover, being socially connected to others in a way that allows one to be seen and valued as a member of the community facilitates thriving and connects to social wellness [20]. Along with social wellness, the Black holistic wellness model is composed of seven dimensions, including cultural wellness, occupational wellness, spiritual wellness, physical wellness, intellectual wellness, and emotional wellness [20].

Previous research on AFAM demonstrates the ways this network operates as a space of cultural health across several dimensions of Black holistic wellness. For example, qualitative interviews highlight how AFAM supports intellectual wellness; specifically, students described AFAM as a space of intellectual stimulation, where they could apply and critique the knowledge they were learning in their courses [21]. Moreover, participation in the network has been associated with positive retention and graduation outcomes, where academic wellbeing is also part of intellectual wellness [20,22–24].

Counterspaces such as AFAM in which students can feel affirmed, validated, whole, and human are necessary. Solórzano et al. [25] define counterspaces as those that counter dominant narratives operating to invalidate and marginalize people of color. Rather than being stereotyped and disregarded, in counterspaces, the experiences, cultures, and identities of people of color are centered, validated, and affirmed. Validating counterstories and narratives of the Black experience contributes to cultural wellness, where Gooden [20] connects aspects of cultural wellness to feeling affirmed in one's racial group. AFAM operated as a space of affirmation and validation for students who were in the network in high school and in college [21,26].

Therapeutic outcomes in AFAM such as safety, universality, vicarious learning, and social learning all connect to social wellness [27]. Social wellness emphasizes "social connection and communalism, or being connected with and to community" [20] (p. 11). Both quantitative and qualitative studies illustrate the ways in which AFAM contributes to social wellness. Across qualitative interviews, students described AFAM as a space of connected-

ness [21], and Grier-Reed and Wilson [24] found that AFAM students tended to be more connected on campus than non-AFAM Black students. In response to identifying whom they turned to when faced with a problem, AFAM students identified more people than their non-AFAM counterparts [24]. Moreover, these students identified more Black people and more university connections in their networks [24]. In sum, students participating in AFAM seemed to have more community, an indicator of social wellness.

The therapeutic factors of catharsis and corrective emotional experiences in AFAM can be connected to emotional wellness [27]. Gooden [20] links emotional wellness to emotional vitality, emotional development, and resilience. Resilience was identified as an important theme in AFAM as students described the network as helping them get through the week or refuel [21]. Having opportunities to vent or achieve catharsis and be validated was part of the experience. Validation of students' experiences, perspectives, and emotions countered the invalidation experienced elsewhere and provided a corrective emotional experience that contributed to emotional wellness [27].

With implications for countering antiblackness in higher education and institutionalizing support for cultural health and wellness, we center and document experiences of antiblackness in the United States through students' stories in AFAM. Specifically, we ask: What were the experiences of antiblackness for students in AFAM? Utilizing notes from discussions in AFAM over a 13-year time frame, we illuminate the ways students experienced antiblackness inside and outside of the education at their PWI. By uplifting these counterstories, we hope to inform and inspire others to take up the call of addressing antiblackness in higher education by creating spaces of cultural health.

## 2. Materials and Methods

### 2.1. Settings

AFAM was developed at a PWI in the Midwestern United States. The PWI serves 48,120 students, 28,747 of these students being undergraduates. Of the undergraduate population, racial demographics consisted of <1% Native American, 4% Black, 8% Asian, 4% Latinx, <1% Hawaiian, 64% White, 3% Multiracial, 12% International, and 4% unknown. The gender breakdown of the university was 53% women and 46% men.

Developed by two faculty members, AFAM was a weekly networking group that met during lunchtime. Weekly meetings included up to 30 students, and there was a constant influx of new faces. AFAM was co-facilitated by Black faculty and graduate students. The initial purpose of AFAM was to create community and to support Black students' retention and graduation at their PWI. The AFAM network included Black university faculty, staff, and graduate students and provided space for undergraduates to engage with each other intellectually, socially, and personally. In the initial years, AFAM was advertised via flyers around campus. As time went on, students tended to find out about the network through word-of-mouth.

Meetings were semi-structured. Because attendance was voluntary, there was always the possibility of new faces, so meetings began with introductions and check-ins where everyone, including facilitators, shared a high and low moment from the week. After the introduction and check-ins, the space was opened for discussion. Students freely shared topics for discussion and directed the conversation each week. Topics ranged from popular culture and celebrities to news events to personal struggles and racial incidents on campus. Once topics were identified by students, facilitators worked to generate meaningful discussion by relying on active listening and basic counseling skills. For a fuller discussion of facilitation skills in AFAM see Grier-Reed et al. (2020) [28] and Grier-Reed and Ajayi (2019) [15].

Participation in the network has been associated with positive outcomes including a higher level of social integration on campus and better retention and graduation outcomes [22–24]. In addition, the group has been found to be therapeutic, providing safety, validation, connectedness, intellectual stimulation, resilience, empowerment, and a home base on campus in ways that overlap with traditional therapeutic factors in group

work [21,27]. Finally, AFAM has been identified as a sanctuary for coping with racial microaggressions and a space for working through racialized labor and combatting racial battle fatigue [14,29].

*2.2. Participants*

From 2005–2006 to 2017–2018, the total number of participants in AFAM was 790. AFAM participants ranged from African American descendants of slaves (ADOS) to first- and second-generation immigrants whose backgrounds included but were not limited to Ghanaian, Ethiopian, Nigerian, Liberian, Cameroonian, Somali, Kenyan, and Eritrean. AFAM participants were undergraduate students across different years in school (i.e., first-year, sophomore year, junior year, and senior year) and came from different programs within the university (e.g., biological sciences, education, liberal arts, business, agricultural and natural sciences, computer science, and engineering). Of these, 70% identified as women, 27% identified as men, and 3% did not report their gender.

Over the span of 13 years from 2005–2006 to 2017–2018, there were a total of 354 AFAM meetings, and notes from 277 (78%) of those meetings were obtained for this study. The remaining notes were missing. Moreover, because AFAM was a group in which students attended voluntarily, no two meetings consisted of exactly the same composition of students. On average, students came to 8.4 meetings, and with a standard deviation of 13, there was significant variation in the number of meetings students attended. Therefore, even though our dataset spans 13 years of time, we do not follow individuals for 13 years. Rather, each AFAM meeting is a slightly different cross-section or composition of students.

*2.3. Procedures*

This study was conducted ethically and with the approval of the institutional review board at the participating university. Group discussion synthesized via the AFAM facilitator note was the unit of analysis, where graduate students who co-facilitated AFAM were assigned to take notes that captured the discussion each week. There was no template for taking notes, and facilitator notes varied with respect to detail and length (typically ranging from one paragraph to $\frac{3}{4}$ of a page). Notes were submitted to the principal investigator (PI) of this study, who is also the first author; she co-founded AFAM and supervised all graduate student co-facilitators.

The research team began by reading and reflecting on critical race theory, where reflexivity was important, and each member of the team drafted a statement of her own positionality and expectations relative to the research. The team was composed of three women—one African American woman and two White women. The African American woman was the PI. She co-founded AFAM and co-facilitated AFAM meetings; she was also a professor in Family Social Science. The two White women on the research team were doctoral students in Family Social Science who did not participate in AFAM. To address the potential power differentials between the professor and graduate students, the PI operated from a feminist perspective to create an egalitarian atmosphere, wherein, " . . . feminists and egalitarians . . . notice and resist the institutionalized and illegitimate status hierarchy . . . " [30]. Sharing power was important across the professor and graduate students and operationalized by actions such as rotating who spoke first during the consensual coding process. Moreover, having the ratio of two students who were AFAM outsiders to one professor who was an AFAM insider helped to ensure voice across different perspectives. Background readings for the research team also included literature related to racial trauma, specifically the work of Comas-Diaz [10]. All research team members noted that they expected to find racism in the data prior to reviewing the AFAM facilitator notes.

The research team adapted consensual qualitative research (CQR) methods to analyze the data [31,32]. CQR is a team-based qualitative method that includes peer examination or debriefing to create trustworthiness in the process. Auditing is also important to the research process, which hinges on finding consensus. The auditor in this study was an

African American man who co-facilitated AFAM meetings during the 2018–2019 academic year; he was also a master's student in the Family Social Science program.

CQR was originally designed for intensive study of 8–15 cases or individuals using interview, questionnaire, or survey data [32]. The method involves teams of 3–5 researchers who, through consensus, determine domains and core ideas within and across cases, and through cross-case analysis identify categories spanning the data [31,32]. The method begins with an individual review of the data to develop a start list of domains or topic areas and then proceeds to individual coding. Once individual coding is complete, researchers meet together to develop consensus. Operating from a feminist perspective that reduces hierarchy and emphasizes the empowerment of all team members, Hill et al. [31,32] emphasize the need for shared power and voice among members of the research team to inhibit groupthink. The authors [31,32] also recommend identifying expectations and biases prior to the start of the research and including an auditor throughout the study. Finally, in reporting results the authors recommend using labels (e.g., general, typical, rare, and variant) rather than reporting numbers or frequencies.

We adapted CQR to examine our dataset of 277 facilitator notes, where in line with recommendations, a team of three researchers worked to analyze the data, and the research included an auditor. The research team began by identifying what they expected to find and reflecting on their personal identities (race, class, gender, sexual orientation, etc.) as suggested by Hill et al. [31,32]. Furthermore, in line with Hill et al.'s [31,32] recommendation to familiarize oneself with the literature, the team engaged in readings of critical race theory and racial trauma to prepare for the study [31,32]. Then, each team member individually reviewed all 277 facilitator notes. Afterward, the team met together to identify a start list of domains derived from their initial reading of the data. The start list of domains and initial coding scheme are shown in Table 1. Once the team agreed on how to code the initial domains, each member individually coded data (using Table 1) by re-reading all 277 notes. Once the individual coding was complete, the research team met together to triangulate their perspectives and find consensus. Data about which there were disagreements were put on hold or in the "parking lot". At the end of the analysis, there were no data left in the "parking lot", and the "Other" domain in Table 1 was recast as "Systemic Racism". The analysis was then turned over to the auditor for review.

**Table 1.** Initial coding scheme.

| Domain | Description |
| --- | --- |
| Racial trauma A (RTA) | Events of danger (experienced, witnessed, communicated) |
| Racial trauma B (RTB) | Threats of harm (experienced, witnessed, communicated) |
| Racial trauma C (RTC) | Shaming and humiliating (experienced, witnessed, communicated) |
| Racial rejection (RR) | Not belonging, feeling excluded, lack of community, institutionally unwelcomed |
| Racial microaggression (RM) | Subtle, ambiguous events of racism (leaves a question mark: Was that racist?)—microinvalidations, microassaults, microinsults |
| Other (O) | Racially adverse experiences which do not fit any other categories 1. Systemic racism |

The auditor on the research project evaluated the team's consensual analysis by reading all 277 AFAM discussion notes in their original form and reviewing and providing feedback on the research team's consensually coded data. The audit included determining whether the raw data were in the correct domains and whether ideas were faithfully represented. Upon completion of the audit, the research team reviewed the feedback and met with the auditor to discuss findings. The analysis was confirmed by the audit, and the domains and themes were finalized in the meeting with the auditor, where in line with recommendations by Hill et al. [31,32] who suggest that similar domains be combined, the

racial trauma (A), racial trauma (B), and racial trauma (C) domains were collapsed into one domain labeled Racial Trauma. We deviated from CQR, however, in reporting our data, where we included frequencies or numbers rather than the labels of general, typical, variant, or rare. Given that we reviewed hundreds of facilitator notes, we thought that numbers or frequencies might be more informative.

## 3. Results

Across the 277 AFAM facilitator notes, we identified discussions of antiblackness in over half (n = 143). Discussions that included aspects of racial trauma occurred in 51 AFAM meetings, followed by discussions that included aspects of racial rejection occurring in 34 meetings. Discussions of racial microaggressions were documented in 33 AFAM meetings, and discussions of systemic racism occurred across 25 meetings. Refer to Table 2 for a summary.

**Table 2.** Summary of results.

| Domains of Antiblackness | Themes |
| --- | --- |
| Racial trauma (n = 51 meetings) | Events of danger including news events such as the Jamar Clark verdict and the Muslim ban as well as violence such as shootings (on campus or in the community) along with police interactions and harassment. Racial trauma also involved threats of danger including a heightened sense of threat with the 2016 election and an uprise in racist and Islamophobic events. Finally, racial trauma involved shame and humiliation associated with being called the n-word, being criminalized, and being oversexualized, made to feel stupid, degraded, dirty, and marginalized. |
| Racial rejection (n = 34 meetings) | Not belonging or feeling excluded and unwelcome in White spaces, including feeling institutionally unwelcomed, where Black students' history, experiences, and identities were not supported or represented on campus or valued in U.S. society. |
| Racial microaggressions (n = 33 meetings) | Cultural insensitivity, where students felt silenced, invisible, invalidated, stereotyped, condescended, and treated coldly, including petty insults by instructors. |
| Systemic racism (n = 25 meetings) | Experiences and perceptions of educational and economic disenfranchisement, cultural appropriation, and the criminalization of Blackness perpetuated by media. |

Note: Two other domains, Racialized Labor and Racial Battle Fatigue, were also identified and previously published [14].

### 3.1. Racial Trauma

Racial trauma involved events of danger, including news events such as the Jamar Clark verdict and the Muslim ban discussed in 2017, as well as violence such as shootings (on campus or in the community) along with police interactions and harassment. Racial trauma also involved threats of danger, including a heightened sense of threat with the 2016 election and an uprise in racist and Islamophobic events. Finally, racial trauma involved shame and humiliation associated with being called the n-word, being criminalized, and being oversexualized, made to feel stupid, degraded, dirty, and marginalized.

Many of the students' group discussions explored current events that were racially traumatic. The following excerpt from one facilitator's note captured one such group discussion where many students communicated their " . . . frustration that a statement was released by the police chief asking people not to riot. Students were unsure if the statement said not to riot or protest. They brought up the differential responses to White riots over sports and Black riots over injustice." The Muslim ban was another current event that came up in group discussions. As noted in the following excerpt, students shared " . . . the scary things that the executive order communicates, such as, devaluing the lives of individuals who hold certain social identities."

Threats and events of danger were part of the racial trauma, including police harassment. In one discussion, the facilitator noted that students shared " . . . reactions to the most recent police-involved shooting of two unarmed Black men in the preceding week.



Students reported feeling sadness and anger. Notably, two students reported fearing for their lives in the hands of police as Black people." In these group discussions, students also shared police-related incidents that occurred on campus. Students described recent events that were hosted by Black student organizations that resulted in the police being called to the scene. The following excerpt from one facilitator's note provides one student's experience at an event as they shared " . . . that police officers from a number of agencies across the Metro region responded with pepper spray, with unnecessary yelling and taunting, and a number of other tactics" which led to further discussions about a similar police response to other Black student events. Another excerpt from a facilitator note captured the students feeling " . . . like these actions were disproportionate and discriminatory", especially when compared to events hosted by predominantly White organizations on campus.

Racially traumatic events in the classroom centered on shame and humiliation. Students shared several incidents in which they heard the "N-word in class", and in one group, a student shared their experience in which a White student used the "N-word as part of her presentation". As previously reported by Grier-Reed and colleagues [4] (p. 77), "students discussed how hearing the word can be traumatic for some, even if the intent of the speaker is not malicious." In the facilitator notes analyzed for this study, one student shared " . . . how it crushed her, went straight to her heart."

*3.2. Racial Microaggressions*

Racial microaggressions focused on cultural insensitivity, where students felt silenced, invisible, invalidated, stereotyped, condescended, and treated coldly by instructors or students. AFAM participants reported on incidents of racial microaggressions that took place in various contexts, such as within classrooms and in social settings. The incidents of microaggressions reported in AFAM included ambiguous actions and subtle verbal affronts from others.

Microaggressions around campus were common occurrences and frequent topics of discussion for AFAM participants. Throughout several instances in the AFAM notes, students reported " . . . microaggressions and petty insults by instructors and staff members on . . . campus." In the following note, one student described how they were consistently confused with another student of color: "One student brought up how a teacher had called her and another student the wrong name several times throughout the semester." The instructor's lack of attempt to distinguish one student of color from another led the AFAM participant to feel invalidated in class. In more egregious examples of classroom microaggressions, " . . . students talked about professors and staff using the N-word."

Within the context of academic spaces, faculty and staff were not the only perpetrators of microaggressions. In many instances, students reported on the additional microaggressions that they faced from peers. In one AFAM note, the facilitator describes a " . . . discussion about White students making assumptions about Black students' competence, being treated coldly, and mistreated by White students." In another instance, one AFAM participant describes experiencing a White student overlooking the Black student's presence. Unsure about the nature of the encounter, the student cried when retelling the story, where she:

> "[D]iscussed an incident of disrespect in a classroom, where a student moved her belongings from the table she was sitting in, leaving her to find another seat before an exam . . . she believes the action was likely related to her identity as a Black woman."

Microaggressions around campus also occurred outside of the classroom. Many students in AFAM reported on their experiences with microaggressions coming from other students in social settings. In the following instance:

> "A student introduced an incident that occurred to him the preceding week, wherein he felt racially microaggressed at a party when asked why he was at the

party. Other students discussed their own experiences with microaggressions in social settings and also discussed the toll of such indignities."

In many cases, participants in AFAM related to not knowing whether or not they experienced a microaggression. Because of the covert nature of microaggressions, students were often left wondering, feeling insecure, or frustrated at the ambiguity of these instances. For example, one AFAM note describes how:

> "The student shared that her roommate's comments about her music choice felt like a microaggression. She shared that she was listening to country music and her roommate commented with surprise at the music choice. Many affirmed this response through their own examples of microaggressions that were often hard to articulate at the moment they happened."

Although most microaggressions reported by students were difficult to identify, some were very explicit. In the AFAM discussion note below, the facilitator recounts an off-campus incident regarding a culturally insensitive party theme:

> "[A] student shared a story about how another predominantly White student group on campus was putting together a 'Black 'Lights' Matter' party but changed it because students were offended. Students explored this issue concluding that it was making light of or mocking an important issue."

### 3.3. Racial Rejection

Racial rejection was associated with not belonging or feeling excluded and unwelcome in White spaces, including feeling institutionally unwelcomed, where Black students' history, experiences, and identities were not supported or represented on campus or valued in U.S. society. Instances of racial rejection occurred most frequently within the context of Black students' relationships with the university. Racial rejection incidents also transpired within the interpersonal relationships between Black students and other students of color, as well as with White students.

In describing their experiences on campus, many AFAM participants reported feeling like they did not belong. AFAM participants cited feelings of institutional estrangement when navigating their university campus, classrooms, and relationships. In the following quote, one student describes the sensation that they felt when contemplating their sense of belonging:

> "[It] can sometimes be like trying to belong to a club that you don't fit into or doesn't really want you to be a part of it . . . A number of students indicated that there have been times in this setting that they didn't feel like they belonged here or experienced feelings of not being wanted here."

In addition to students feeling personally unwelcomed on their campus, they also reported the feeling that their culture at large was being downplayed or dismissed altogether. One AFAM note details how students recognized that " . . . dormitories are downplaying Black History (BH) month. The group picked up where it left off with the issue of BH month being phased out by other monthly follies such as 'feelings month' or recycling." In other notes, students continued their conversations of unbelonging when discussing the university's alleged commitment to its diversity initiatives. Students frequently described "feeling 'used' for a diversity quota on campus and feeling isolated, not a part of the University" as reported in a previous study of AFAM by Grier-Reed et al. [28] (p. 12). Further elaborating on the disconnect between the university's stated goals and its realities, students discussed how they felt the university feigned its commitment to diversity by saying:

> "The only thing diverse about the U are its brochures. A number of students reported that they felt misled by the U's marketing. The university wants diversity because it will increase the university's status not because it's a human right. The U wants to be a world class university but won't do things to bring in people of

color. It's so bad that you get excited when you see people of color. The U should make more efforts to recruit people of color."

Additionally, students at the university expressed dismay at how they perceived their history and culture were being glossed over within university classrooms and subsequently misrepresented in the broader society. This sentiment is highlighted by the following quote:

> "Students shared frustration with the bias of history and social science classes, where the horrors of slavery tend to be minimized along with the contributions of Blacks to the United States. They pondered whether education about the Holocaust in Germany was treated the same way. This conversation led to stereotypes about Blacks and Black youth in particular, including the overarching messages we get from the larger society about our identity, who we are supposed to be, and social position."

Although many conversations in AFAM centered around students' feelings and perceptions of the university, these conversations did not happen in a vacuum. Many times, the discourse was embedded in a larger conversation regarding current events, most of which involved the university.

In several conversations, students reported on their feelings regarding the then-recent closure of a university college, which was home to a large proportion of students of color. In addition to the college's dissolution, the Afro Studies department was also facing fiscal cuts, and the university was attempting to scale back the involvement of a multicultural center in new student orientation activities. All of these happenings are reflected in the following quote:

> "Students responded negatively ... as if they were being swept under the rug. Students related this moving on with being more disenfranchised and made linkages between the loss of the ... College and attempts to reduce and/or remove MCAE [Multicultural Center for Academic Excellence] from orientation, along with the continual even if gradual disenfranchisement of the Afro Studies Dept. How orphaned students felt at the university without a place or space to exercise their voices became clear. What happened to the ... College, MCAE, and Afro Studies all seemed to represent symbols of the university's attitude toward African Americans."

In addition to the myriad ways students felt unwelcomed by the university itself, students also reported feeling unwelcomed by other Black-identified students around campus. Racial rejection was apparent in the students' descriptions of their interactions with other Black students, specifically between students who identified as African American and African immigrant students, as exemplified by this quote:

> "One student described the disparaging comments he received from another Black colleague around his African identity. This spurred further conversation regarding tensions between the African American community and those in the African community. Many of the students agreed that they've experienced some conflict or heard comments regarding the other ethnic group. A couple of the group members shared that when they immigrated to the US the group that sponsored them spoke of the need to avoid African Americans. Others talked about the messages they received from parents around dating and socializing that shaped the way they think about the other group."

*3.4. Systemic Racism*

Systemic racism included students' experiences and perceptions of educational and economic disenfranchisement, cultural appropriation, and the criminalization of Blackness perpetuated by the media. Several group discussions encapsulated students' thoughts and feelings as they shared examples of these experiences. Highlighting antiblackness in society today, the conversations illuminated the ways it is lived out in social and institutional contexts.

Experiences in the educational system as a Black student were the topic of many group discussions. In one meeting note, students shared how "Black history isn't taught in schools" and how, as reported by Grier-Reed and colleagues [4] (p. 75), this indicated the " . . . close association between [W]hiteness and education because Blackness is suppressed." On the topic of high school, students shared the differences between suburban and inner-city schools, including preparation for college and different learning styles. In the discussion about the ACT or the American College Test used for admissions, the following excerpt identifies the messages they received about the standardized test.

> "[S]ome high schools tell their students that if they get an 18 on their ACT, that would be good enough to be considered for college—well below the average of admissions . . . setting students up for failure . . . suburban schools encourage students to take the ACT 2 to 3 times. College prep standards. Not the case in a lot of the urban schools."

When it came to their experiences at college, many students expressed frustration at the lack of support or helpful feedback from instructors. This was noted in the following excerpt from a group discussion with many students sharing that " . . . despite their hard work, seeking feedback, and attending office hours they were still struggling to attain passing grades." One student expressed her frustrations " . . . with the way she received feedback that was often unclear, untimely, and graded more harshly than her peers."

There were several group discussions regarding campus safety and the racial descriptions used in the crime alerts sent to the campus community. A facilitator noted that in one group discussion, students also reflected on whether " . . . what you look like dictate[s] how you are treated" after a riot occurred on campus. In another discussion, the facilitator noted that there was conversation related to accountability, and students shared that " . . . most rioters tended to be White . . . White students tend to have more leeway on campus."

This topic came up in another group discussion as students discussed the charges for the Minneapolis police officer, Mohamed Noor. In the following excerpt, students discussed the charges against the officer in comparison to other officers charged with similar crimes.

> "Students discussed the intersectionality of Noor being a part of the Black community and Diamond being White. Many felt as though the charge was not in line with other police involved shootings, and race was a critical factor in deciding to charge with murder. Many felt this charge was expected as 'this is a White Country, we are living in their land' and that this highlights the 'us vs. them' mentality inherent with the US. The conversation then changed to the Parkland shooting response to students compared to students from Chicago and other places who have come out against gun violence. Students again felt as though the disparate response was based on White fear of Black and brown communities."

## 4. Discussion

We examined how students in the African American Student Network (AFAM) experienced antiblackness by analyzing notes from their weekly discussions over a 13-year period of time (from 2005–2006 to 2017–2018); our results centered on racial trauma, racial microaggressions, racial rejection, and systemic racism (see Table 2). Notably, in AFAM as in the larger society, the police force, including campus police, were a major source of racial trauma and a venue for systemic racism. Racial trauma (found in 51 AFAM discussion notes) was the primary way antiblackness was documented, followed by racial rejection, racial microaggressions, and systemic racism, respectively. We incorporated the Comas-Diaz [10] definition of racial trauma that included experiences of shame and humiliation as well as threats and/or events of danger. Furthermore, racial trauma included direct experiences as well as observed experiences of shame, humiliation, and danger. Many racially traumatic events were shared through stories, including personal experiences, viral videos, and news reports.

### 4.1. Antiblackness, Racial Trauma, Microaggressions, and Mental Health

While there is substantial literature focused on racial microaggressions [25,33,34], the research literature on racial trauma is still emerging, and our study suggests that it may be a prominent way Black students at a PWI experience antiblackness, especially given the racial divisiveness that occurred in the 2016 election and that has continued over the last four years. Resulting from the experience or witnessing of racism, race-based stressors, and/or discrimination, racial trauma is similar to post-traumatic stress disorder in symptomology, including symptoms such as flashbacks, helplessness, avoidance, difficulty concentrating, and hypervigilance. Nonetheless, racial trauma differs due to the continued exposure and re-exposure of race-based stressors impacting both the individual and the community. The ongoing experiencing of cumulative racial trauma and microaggressions can affect both mental and physical health [35]. Moreover, the intergenerational effects of slavery, Jim Crow, and systemic racism indicate that the impact goes beyond the individual to the community as a whole.

### 4.2. Antiblackness, Systemic Racism, Racial Rejection, and Mental Health

Healing from racial trauma can be complicated because it occurs on a continuous basis, and Eurocentrism in mental health contributes to the lack of an official diagnosis for racial trauma and culturally relevant treatment interventions. Eurocentrism refers to the systemic racial and cultural bias embedded in conceptualizations of mental health, which are fundamentally rooted in White middle-class value systems. The systemic centering of Whiteness in mental healthcare operates to marginalize the needs and experiences of Black people, including their history, culture, and lived experiences. At its most fundamental level, the centering Whiteness or White institutional presence [9] can be experienced as a rejection of Blackness, i.e., racial rejection and antiblackness. Moreover, as illustrated throughout the history of higher education, the centering of Whiteness and the racial rejection of Black people are deeply entrenched across predominantly White institutions and historical eras.

### 4.3. Implications for Cultural Health

By centering the experiences of Black students in AFAM over a 13-year time period, we bring attention to the ongoing nature of racial trauma and discuss possibilities for cultural health. In contrast to the more individualistic and intrapsychic focus of mental health, cultural health focuses on understanding the individual in the larger context of their community/culture and pushes back against the cultural discontinuity that has also been found to be related to depression and negative self-esteem for students of color [36]. Our research and scholarship have implications for moving away from antiblackness and toward cultural health in predominantly White environments by addressing culture and the cultural incongruence embedded in rugged individualism and monoculturalism.

### 4.4. Addressing Culture and Mental Health

Scholars have suggested that the individual psychotherapy framework may be inappropriate for students of color as focusing on one-to-one counseling does not fully consider the person in the context of their environment [37]; this may be one reason why Black and Latino communities tend to underutilize professional mental health services [38]. Moreover, the interracial context of counseling and therapy represents a microcosm of the larger society, wherein students in our study experienced racial trauma, racial rejection, racial microaggressions, and systemic racism [39–41]. However, culturally responsive practices have been connected to positive academic and psychological outcomes for students. Museus [42] found that organizations with an ethnic focus facilitated cultural adjustment and a sense of belonging for minority students by serving as sources of cultural familiarity, vehicles for cultural expression, and venues for cultural validation. This echoes the work of Hope, Hoggard, and Thomas [43].

### 4.5. Addressing Cultural Incongruence and PWIs

Lacking robust institutional support for cultural identities, PWIs may not fully address the needs of students of color, where for many, the college experience requires a period of transition and adaptation that involves coping with a plethora of problems related to their educational and mental wellbeing [44]. Furthermore, for students of color, the transition may be more burdensome than for their White peers because, for many, they are navigating an institution of higher education with which they have had little interaction beforehand [45]. To ease this transition into higher education, increased attention should be paid to the expectations and values of PWIs, particularly expectations that are incongruent with the values and behaviors of students of color [46].

The mismatch between PWIs and students of color is best explained by cultural incongruence. An extension of cultural discontinuity, the concept of cultural incongruence posits that there is a lack of recognition and a devaluing of the cultural assets that students of color bring to college [47]. This cultural incongruence contributes to experiences of racial rejection and cultural discontinuity that can impact academic performance and mental health. In contrast, an investment in cultural health that values Blackness divests from the Eurocentric bias that is endemic to the culture of higher education and academia at large [36,48].

### 4.6. Pushing Back against Individualism

Much like the mental health system, Eurocentric PWI value systems in higher education are based in individualism in contrast to the more collectivist orientations of many people of color. Whereas many western cultures tend to value individualism, non-western cultures (African, Asian, Latin American, Native American) tend to emphasize collectivism [49]. Moreover, research suggests that collectivist-oriented students face the risk of academic underachievement when they fail to assimilate to individualist norms [48,50,51]. Addressing the needs of collectivist-oriented students is a crucial component of facilitating success among college students of color [52].

Institutionalizing support for cultural health is one important way to shift away from rugged individualism. Like Utsey et al. [19], we underscore the need for culture-specific coping methods that originate from an African-centered worldview. Coping strategies in line with this worldview may serve as a buffer against anxiety, depression, and hostility [17]. Blackmon et al. [16] conceptualized these coping strategies as "africultural coping", which is defined as "culture-specific coping practices used by African Americans" [16] (p. 550). Informal networks and spaces of support, like AFAM, can be considered culturally appropriate and regarded as a preferred way of coping for African Americans [15,27].

### 4.7. Pushing Back against Monoculturalism

Our focus on cultural health advocates for institutional spaces that support students in developing a sense of pride and resilience in their cultural backgrounds, which is not a given due to the prolific nature of monoculturalism and antiblackness in education. As previously discussed, institutions of higher education, specifically PWIs, are embedded in a White racial frame [3,4], which propagates monoculturalism throughout various aspects of the college experience [9]. Students of color experience this monoculturalism as the dominance of White standards, which creates an external pressure to assimilate or "fit in" [4] (p. 67) to those standards or be rejected, degraded, and penalized academically [48, 50,51]. This results in cultural discontinuity, dissonance, and negative experiences for students of color. Investing in institutional resources that facilitate cultural health can push back against monoculturalism and antiblackness by valuing the cultures of Black students and students of color and by building communities in which students can generate a wellspring of pride and resilience in their cultural backgrounds.

## 5. Limitations

Although this research included prolonged engagement, reflexivity, thick description, triangulation, and auditing, the limitations should be acknowledged. For example, the unit of analysis was facilitators' notes rather than actual transcripts of group discussions. With sensitivity to the antiblack context and issues of trust, audio recording did not seem feasible. Weighing the risks of eroding trust, personal sharing, and comfort due to fear of being surveilled in what was purportedly a safe space, the PI relied on facilitators to take notes to capture discussion instead. Although having facilitators take notes is potentially less threatening for students than audio recording group discussions, such notes are also less accurate and less detailed. Moreover, because there was no standard template for notetaking across the 13 years, there was also significant variation in facilitators' notes, where the interpretive biases of the facilitators may also be a significant limitation. Nonetheless, one strength of the open notetaking is that we also avoided a priori categories that primed facilitators to tune into some data while inevitability tuning out others. Instead, facilitators were tasked with capturing the discussion to the best of their ability. In other words, the facilitators (who were supervised by the PI) used their judgement to capture the essence of what was said in group each week. Importantly, given that we relied on facilitator notes rather than transcripts of students' own words, the inclusion of member checking would have strengthened the credibility of our findings. Moreover, even though we found examples of antiblackness in over half of our discussion notes (n = 143), triangulating additional data sources such as observations or interviews could have strengthened the dependability of our results.

## 6. Conclusions

In our study, students in the 21st century experienced antiblackness, primarily in terms of racial trauma, racial rejection, racial microaggressions, and systemic racism, and we assert that one antidote to antiblackness in higher education is a concerted institutional shift toward promoting the cultural health of Black people. We use the term cultural health to emphasize the need for institutionalized spaces that support Black students in developing and maintaining a sense of pride and resilience in their cultural identities. Our focus on cultural health contrasts with the more individualistic notion of mental health, which may be culturally incongruent with more collectivist value systems. Creating spaces of cultural health, such as an Institute for Cultural Health that includes cultural resource centers, better aligns with emic perspectives and Black holistic wellness [20].

The focus on culture highlights the importance of community and context, where in marginalized communities, researchers have found that "well-being was inextricably tied to being able to express themselves not only without fear of rejection but also with the expectation of being embraced and accepted" [53] (p. 1162). Hence, cultural resource centers staffed by those with shared cultural identities can be an avenue for resource-rich informal networks that better align with help-seeking behaviors for people of color and in which they can expect to be embraced, especially when compared to seeking out professional counseling [16,53,54]. Moreover, as Hudson and Romanelli [53] point out, having a shared identity can uniquely equip community members to help each other, highlighting "[t]he health-promoting community characteristics of interconnectedness . . . [based in] . . . a sense of acceptance and support" [53] (p. 1162).

**Author Contributions:** Conceptualization of article, T.G.-R., R.S., and M.Q.; research design, data collection and analysis, T.G.-R.; original draft preparation, T.G.-R., R.S., and M.Q.; review and editing, T.G.-R., R.S., and M.Q. All authors have read and agreed to the published version of the manuscript.

**Funding:** This research was funded by NIFA Hatch Project No. MIN-52-097.

**Institutional Review Board Statement:** The study was conducted according to the guidelines of the Declaration of Helsinki, and approved by the Institutional Review Board of the University of Minnesota (protocol #1005S81858 originally approved 6/22/10 with updated protocol approved 7/14/17).

**Informed Consent Statement:** Informed consent was obtained from all subjects involved in the study.

**Data Availability Statement:** The data are not publicly available due to privacy and confidentiality.

**Acknowledgments:** We would like to acknowledge Alyssa Maples, Anne Williams-Wengerd, and Demitri McGee for their contribution to the analysis.

**Conflicts of Interest:** The authors declare no conflict of interest.

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
