# Peer review of "From Antiblackness to Cultural Health in Higher Education"

_education, doi:10.3390/educsci11020057_

Round 1
Reviewer 1 Report
This manuscript addresses the topic of racism in higher education, specifically antiblackness, and reports rich findings from a qualitative in-depth study. Therefore, in my opinion, the manuscript has great potential to enrich the contemporary discourse in educational science. However, I have some significant concerns and remarks which I think need to be addressed in the revised manuscript:
(1) Overall, the perspective of the manuscript seems to me to be too US-specific to be published in an international journal. What is missing here is a clarification of why the present study is also relevant and important for non-US readers and to what extent the results of this study might be transferable to other (non-US) contexts. Therefore, I recommend that the authors align their argumentation much more with an international audience.
(2) There is no explicit research question stated within the manuscript. The research questions should definitely be added and the manuscript's leitmotif should be adjusted accordingly.
(3) As a reader, I wonder why the authors did not analyze their data longitudinally, as it was collected over a very long period of time (13 years). Therefore, I would appreciate some clarification of this question in the revised manuscript.
Reviewer 2 Report
The main idea of the manuscript is quite interesting, I have learned from it. Despite this, the structure and method of this is not like an article, it seems more like a book chapter. There are some important parts, such as those related to data analysis, numbers and method that are not clear and make it difficult to understand the research carried out.
Some comments:
Abstract: More information about the methodology “using an adaptation of consensual qualitative research methods” What methods are adapted? What kind of adaptation?
Introduction:
1.4 The African American Student Network (AFAM) where the current research took place can be a source of inspiration for institutionalizing cultural support.
Not clear why AFAM can be that: some reasons should be given to defende this idea. AFAM is not already that? Why don't start defining AFAM actions and then these actions could be the insiparion reasons.
Method / 2.1 Settings: What was the standard structure of a meeting, what kind of intervention produce on students? How many students repeat on various meetyings?. The meeting process is not clear and the way students come to meetings is not explained. Are they convoqued? Do they come by their own interest? This is the basis of the data analysed on the article ans should be quite clear.
Procedures: facilitator note was the unit of analysis, where graduate students who co-facilitated AFAM were assigned to take notes that captured the discussion each week.
Was there a previous criteria, template or any instrument to take these notes and to make posible to compare them or was just a transcription ?
This should be on the participants section: “Over the span of 13 years from 2005-06 to 2017-18, there were a total of 354 AFAM meetings, and notes from 277 (78%) of those meetings were obtained for this study. The remaining notes were missing.”
PI operated from a feminist perspective to create an egalitarian atmosphere: What does feminist perspective mean.
What kind of actions are linked to this perspective. An example could be shown?
Research team’s coding matrix: Thise matrix is not showed and it is very relevant to understand the way data was analysed. Can this matrix be shown?
Table 1. Summary of Domains of Antiblackness: This table does not have any number. Can be shown the % of each domain found on the notes for example?? Just to compare the relevance of each of them.
Our results centered on racial trauma, racial microaggressions, racial rejection, and systemic racisme: ok, why are these the 4 domains... is there any justification for that, maybe the % of presence on the notes. Show those numbers.
Discusion: The ongoing experiencing of cumulative racial trauma and microaggressions can affect both mental and physical health: is there any reference to support this idea?
Round 2
Reviewer 1 Report
Although in my opinion the manuscript is still very U.S.-specific (the international dimension of the topic could be emphasized further), all in all, in my judgment, the revised manuscript is acceptable for publication.
Reviewer 2 Report
The authors have corrected all the proposed changes clearly enough. Now the manuscript is clearer and has a better methodological explanation which gives it more coherence.